# Efficiency of the Generalized Second-Price Auction for Value Maximizers

## ABSTRACT

We study the price of anarchy of the generalized second-price auction where bidders are value maximizers (i.e., autobidders). We show that in general the price of anarchy can be as bad as 0. For comparison, the price of anarchy of running VCG is 1/2 in the autobidding world. We further show a fined-grained price of anarchy with respect to the discount factors (i.e., the ratios of click probabilities between lower slots and the highest slot in each auction) in the generalized second-price auction, which highlights the qualitative relation between the smoothness of the discount factors and the efficiency of the generalized second-price auction.

**ACM Reference Format:**
Anonymous Author(s). 2018. Efficiency of the Generalized Second-Price Auction for Value Maximizers. In *Proceedings of ACM Conference (Conference'17)*. ACM, New York, NY, USA, 11 pages. https://doi.org/XXXXXXX.XXXXXXX

## 1 INTRODUCTION

Generalized second-price (GSP) auctions have been the prevalent auction format deployed in the online advertising market for almost two decades [Edelman et al., 2007, Varian, 2007]. GSP auction, a multi-slot extension of the classic Vickrey auction [Vickrey, 1961], was initially designed for advertisers maximizing (quasi-linear) utility given by the difference between value and payment. There is a large body of literature investigating the theoretical properties of GSP with utility-maximizing advertisers [Aggarwal et al., 2006, Gomes and Sweeney, 2009, Hartline et al., 2014, Lahaie, 2006, Lucier et al., 2012, Nekipelov et al., 2015, Roughgarden et al., 2017, Thompson and Leyton-Brown, 2013]. In particular, Edelman et al. [2007] and Varian [2007] characterized the class of envy free equilibria (a.k.a., symmetric Nash equilibria), and showed that envy free equilibria coincides with the equilibrium of VCG in terms of allocation so that they are always efficient. Caragiannis et al. [2015] characterized its price of anarchy (PoA) [Koutsoupias and Papadimitriou, 1999], the ratio between the worst welfare performance in equilibrium and the socially optimal welfare for both the Bayesian setting as 1/2.927 and the full information setting as 1/1.259.

However, the online advertising market has witnessed a significant shift towards *autobidding* in recent years. Autobidding, namely the procedure of delegating the bidding tasks to automated bidding agents to procure online advertising opportunities, has become a widely adopted mode of bidding, contributing to more than 80% traffic of the online advertising market [Dolan, 2020]. Instead of

setting their bids manually, the autobidding product allows the advertisers to specify their high-level objectives and constraints. With these inputs, the autobidding agents adjust bids in real time to optimize on behalf of the advertisers (see Aggarwal et al. [2019], Balseiro et al. [2021a], Deng et al. [2021] for more background on auto-bidding). A popular autobidding product is target cost-per-acquisition (target CPA), which aims to maximize the total value subject to a target return-on-investment (ROI) constraint that specifies the minimum admissible ratio between value and payment. Such a procedure greatly simplifies the interaction between the advertisers and the ad platform; and moreover, the online autobidding agent can better optimize the advertisers' bids by leveraging advanced machine learning techniques to predict how different bids impact the advertisers' payoffs. Following the recent line of research on autobidding [Balseiro et al., 2021a], in this paper, we denote advertisers maximizing their quasi-linear utility by *utility maximizers* and denote advertisers adopting autobidding products by *value maximizers*.

Such a significant shift in the behavior model of bidding agents raises interesting and important research questions to revisit the effectiveness of the existing mechanisms from the perspective of autobidding [Aggarwal et al., 2019, Balseiro et al., 2021a, Deng et al., 2022, 2021, Liaw et al., 2022]. Aggarwal et al. [2019] show that the PoA of running the second-price auctions with value maximizers is only 1/2 in contrast to the well-known result that running the second-price auctions with utility maximizers is optimal in terms of social welfare. Recently, inspired by the shift from the second-price auctions to the first-price auctions in the display ad markets [Paes Leme et al., 2020], Liaw et al. [2022] and Deng et al. [2022] show that the PoA of running the first-price auctions with value maximizers is also 1/2. However, the PoA of the widely adopted GSP auctions for value maximizers still remains open. In this paper, we aim to *characterize the PoA of running GSP auctions under autobidding*.

### 1.1 Our Results

In this paper, we characterize the PoA of running the GSP auctions for value maximizers. It turns out that the PoA can be as bad as 0 in the worst case. In comparison, the PoA of running VCG (in particular, second price auctions) is $1/2$[1]. However, the worst case instance for PoA being 0 is unlikely to appear in practice as it contains an auction in which the ratio (a.k.a., discount factor) between the second slot and the first slot approaches 0.

Our main contribution in this paper is to provide a fine-grained characterization of PoA based on the sequence of discount factors (Theorem 5.1). Our PoA bounds are tight in the worst-in-class sense, i.e., among all instances that share the same bound as given

---

[1]The PoA being 1/2 for VCG depends on the assumption that the bidders do not adopt dominated strategies. In the case when the bidders can adopt dominated strategies, the PoA of running VCG can be 0. In contrast, PoA can be 0 for GSP even assuming that the bidders do not adopt dominated strategies.

by Theorem 5.1, there is one instance for which the bound is tight (Theorem 6.1). We complement our tight but complex bounds with slightly loose but simplified bounds (Corollary 5.2) that highlight the qualitative relation between the smoothness of the discount factors and the efficiency of the generalized second-price auction: The smoother the discount factors are, the better the PoA bound is. Our bounds are obtained by a novel analysis that (1) establishes the Pareto frontier of welfare decomposition between value and payment for each auction with a charging scheme tailored to the GSP auction, and (2) aggregates the Pareto frontiers across auctions in an optimal way.

## 1.2 Further Related Work

Following the seminal works in GSP auctions for utility maximizers from [Aggarwal et al., 2006, Edelman et al., 2007, Varian, 2007], Lahaie [2006] bounds its PoA for special cases in which click-through rates decay exponentially at a fixed rate; Gomes and Sweeney [2009] characterize the existence of symmetric Nash equilibria in Bayesian settings with i.i.d. distributions. In addition to welfare performance, revenue performance of GSP auctions for utility maximizers has also been extensively studied in Hartline et al. [2014], Lucier et al. [2012], Thompson and Leyton-Brown [2013].

Motivated by the recent rapid adoption of autobidding in online advertising, there is a growing body of literature examining its impact on mechanism design in an autobidding world. In the seminal paper of Aggarwal et al. [2019], they show that uniform bidding is optimal for truthful auctions (with respect to quasi-linear utility maximizers), demonstrate the existence of equilibrium, and prove the price of anarchy results for truthful auctions. Balseiro et al. [2021a, 2022] consider the setting of Bayesian mechanism design and provide the characterization of the revenue-optimal auctions under different information structure and budget constraints. In order to improve the PoA of the system, Mehta [2022] and Liaw et al. [2022] show how to leverage randomization and non-truthfulness to improve welfare efficiency. When machine-learned advice approximating buyers' values is given, using boosts and reserves are provably shown to be effective in improving the welfare efficiency guarantees [Balseiro et al., 2021b, Deng et al., 2022, 2021].

## 2 PRELIMINARIES

*Ad auctions.* Following prior work in autobidding [Aggarwal et al., 2019, Deng et al., 2022, 2021, Liaw et al., 2022], we consider the following multi-auction model: There are $n$ bidders participating in $m$ auctions, where we will generally use $i$ to index bidders, and $j$ to index auctions. Without loss of generality, we assume each auction $j$ has $s$ winning slots (where normally $s < n$), and we generally use $k$ to index such slots. In each auction $j$, each bidder $i$ has a value $v_{i,j}$, and each slot $k$ has a discount factor $d_{j,k}$. The value that bidder $i$ receives when winning slot $k$ in auction $j$ is $v_{i,j} \cdot d_{j,k}$. Without loss of generality, we assume that $d_{j,k} \geq d_{j,k+1}$ for any $j \in [m]$ and $k \in [s-1]$. For notational simplicity, we also assume $d_{j,k} = 0$ for any $k > s$.

*The generalized second-price auction.* We focus on the generalized second-price auction in this paper: Each bidder $i$ submits a single bid $b_{i,j}$ in each auction $j$. Let $i(j,k)$ be the bidder with the $k$-th largest bid in auction $j$.[2] Then for each $j \in [m]$ and $k \in [s]$, bidder $i(j,k)$ wins slot $k$ in auction $j$, receives value $\mathrm{val}_{i(j,k),j} = v_{i(j,k),j} \cdot d_{j,k}$ and pays the $(k+1)$-th largest bid discounted by $d_{j,k}$, i.e., bidder $i(j,k)$ pays $p_{i(j,k),j} = b_{i(j,k+1),j} \cdot d_{j,k}$. We make a few remarks:

- The above fully specifies the slot received and payment made by every bidder $i$, since the mapping $k \mapsto i(j,k)$ gives a permutation of the $n$ bidders $[n]$.
- $\{i(j,k)\}$, $\{\mathrm{val}_{i,j}\}_i$ and $\{p_{i,j}\}_i$ depend on $\{b_{i,j}\}_i$. We will make this dependence explicit as needed.
- In general, bidders may bid randomly, in which case $\{b_{i,j}\}$, $\{i(j,k)\}$, $\{\mathrm{val}_{i,j}\}$ and $\{p_{i,j}\}$ are all random variables. In such cases, $\{b_{i,j}\}$ must be independent across different bidders $i$.

*ROI-constrained value-maximizing bidders.* We consider autobidders, who are technically ROI-constrained value maximizers. That is, each bidder aims to maximize the total value they receive, subject to the constraint that the total payment they make does not exceed the total value. Formally, each bidder $i$ solves the following optimization problem when deciding her (possibly randomized) bidding strategy $\boldsymbol{b}_i$ given all other bidders' bids $\boldsymbol{b}_{-i}$:

$$\max_{\boldsymbol{b}_i} \quad \sum_{j \in [m]} \mathbb{E}_{\boldsymbol{b}_i, \boldsymbol{b}_{-i}} [\mathrm{val}_{i,j}]$$

$$\text{subject to} \quad \sum_{j \in [m]} \mathbb{E}_{\boldsymbol{b}_i, \boldsymbol{b}_{-i}} [\mathrm{val}_{i,j}] \geq \sum_{j} \mathbb{E}_{\boldsymbol{b}_i, \boldsymbol{b}_{-i}} [p_{i,j}].$$

*Equilibria and the price of anarchy.* We study the worst-case efficiency of the generalized second-price auction in equilibrium, as measured by the price of anarchy. In words, the price of anarchy is the ratio between the total value received by all bidders in the worst equilibrium, and the optimal social welfare disregarding incentive issues. Formally, given $n$, $m$, the values $\{v_{i,j}\}$ and discount factors $\{d_{j,k}\}$, let $\mathrm{opt}_j$ be the contribution of auction $j$ to the optimal welfare, i.e.,

$$\mathrm{opt}_j = \sum_{k \in [s]} v_{i^*(j,k),j} \cdot d_{j,k},$$

where $i^*(j,k)$ is the bidder with the $k$-th largest value in auction $j$. We are interested in the worst-case price of anarchy (PoA) over bidders' values, i.e.,

$$\mathrm{PoA}(m, s, \{d_{j,k}\}) =$$

$$\inf_{n, \{v_{i,j}\}, \{\boldsymbol{b}_i\}_i \text{ form an equilibrium}} \frac{\sum_{i \in [n], j \in [m]} \mathbb{E}[\mathrm{val}_{i,j}]}{\sum_{j \in [m]} \mathrm{opt}_j}.$$

## 3 THE ANALYSIS AT A GLANCE

*Limitations of and insights from existing approaches.* First let us review existing approaches of analyzing the PoA bounds under autobidding. Key to many existing methods is the following simple observation: The ROI constraints guarantee that the total amount that all bidders pay is a lower bound of the total value that all bidders receive. So, any lower bound on the total payment is also a lower bound on the total value. Then, to lower bound the total value, one only needs to establish "substituting" lower bounds on

---

[2]For simplicity, we assume ties are broken in favor of the bidder with the smaller index.

the total value and the total payment respectively, and argue that one of the two lower bounds must be large enough.

A simple concrete example is the analysis of the second-price auctions with the no-underbidding assumption (as underbidding is a dominated strategy for value maximizers in the second-price auctions) [Aggarwal et al., 2019, Balseiro et al., 2021b, Deng et al., 2021]: Suppose there is only one slot that matters in each auction, i.e., $d_{j,1} = 1$ and $d_{j,k} = 0$ for all $k > 1$. Moreover, suppose all bidders bid deterministically, and never bid below their values, i.e., $b_{i,j} \geq v_{i,j}$ for all $i$ and $j$. Then the following argument shows the PoA is at least $1/2$: Consider each auction $j$. If the bidder with the highest value (henceforth the rightful winner) in $j$ also has the highest bid, then the rightful winner wins in $j$, and the value they receive is precisely the contribution of $j$ to the optimal welfare. Otherwise, since the rightful winner bids at least their value, the actual winner must pay at least the rightful winner's value, which is the contribution of $j$ to the optimal welfare. So $j$ contributes at least its contribution to the optimal welfare, either to the total value or to the total payment (note that we are relaxing the contribution and considering lower bounds of the two quantities). It is not hard to see that the worst-case situation is when the contributions of all auctions are equally split between the value and the payment, in which case we get $1/2$ of the optimal welfare. It turns out this ratio is tight for the second-price auction. (See proof of Theorem 3.3 in [Deng et al., 2021] for more details).

However, this argument no longer works as underbidding is no longer a dominated strategy for value maximizers in GSP auctions. In order to obtain the tight PoA for the generalized second-price auction, we aim to develop new techniques to establish optimal tradeoffs between value and payment lower bounds.

*Our approach.* We break down the analysis into two parts: (1) establishing the optimal tradeoff between the contributions to the value and the payment of each individual auction, and (2) aggregating over all auctions in the optimal way to obtain the tight worst-case lower bound on the welfare. In the first part, we characterize the Pareto frontier between the contributions to the value and the payment of each auction. This is done by considering which slot each bidder would get if they bid their true value (a *phantom* bidding strategy), fixing other bidders' bids. The value obtained by the bidder under the phantom bidding strategy is denoted by *proxy value*. When aggregated over auctions, the total proxy value each bidder receives under this phantom bidding strategy is a lower bound of that bidder's contribution to the value, since bidding true values in all auctions is a feasible bidding strategy (i.e., the ROI constraint is satisfied). Moreover, such phantom bidding strategies also give lower bounds on the contribution to the payment: For example, if a bidder with value $v$ would get the $k$-th slot by bidding $v$, then the $(k-1)$-th largest bid must be at least $v$, and by the payment rule, the top $k-2$ winners must each pay at least $v$ discounted by the corresponding slot's discount factor. We observe that the tradeoff is dictated by the slot that the bidder with the highest value would win when bidding their true value. This means there are $s+1$ cases that we need to consider, which correspond to the $s+1$ points (one of which is trivial) on the Pareto frontier. We bound the coordinates of each of these points using a charging scheme tailored to the generalized second-price auction (see Fig 1 and 2 for visual demonstrations).

Once we have the Pareto frontiers of all auctions, we can proceed to the second part of the analysis, which is aggregating them in the optimal way. This involves two conceptual steps: characterizing the worst-case distribution of the optimal welfare into individual auctions, and on top of that, determining the worst-case split between the contribution to the value and the contribution to the payment in each auction. From a geometric point of view, both steps boil down to the same operation: taking the convex closure of the optimal tradeoffs obtained in the first part of the analysis. The tight PoA ratio then corresponds to the (lower) intersection point of this convex closure with the line $y = x$. Then, exploiting the structure of the Pareto frontier curves, we obtain an explicit formula for the outcome of the optimal way of aggregation, which gives the tight PoA of the generalized second-price auction (see Fig 3 for visual demonstrations). This is further complemented with hard instances showing each term in our bound is necessary. Our high-level approach can be adapted to other auction formats with local changes, which may be of broader interest.

## 4 BOUNDING THE CONTRIBUTION OF AN INDIVIDUAL AUCTION

In this subsection, we fix any bidding strategies $\boldsymbol{b} = \{b_{i,j}\}$ (not necessarily forming an equilibrium; we will use the equilibrium condition later), and bound the contribution of each auction to the proxy value and payment. In doing so, we will consider the proxy value each bidder $i$ receives in each auction $j$ under the phantom bidding strategy where they bid their true values, i.e., $\text{val}_{i,j}(b_{i,j} = v_{i,j}, \boldsymbol{b}_{-i,j})$. For brevity, we let $\text{pval}_{i,j} = \text{val}_{i,j}(b_{i,j} = v_{i,j}, \boldsymbol{b}_{-i,j})$. Note that $\text{pval}_{i,j}$ is generally a random variable. By default, when we write $\text{val}_{i,j}$ or $p_{i,j}$, they are induced by the fixed bidding strategies $\boldsymbol{b}$. We will replace this proxy value with the actual value received by each bidder later after we aggregate the contributions of all auctions.

The main claim we prove in this subsection is the following lemma, which characterizes the worst-case tradeoff between each auction's contribution to the (proxy) value and its contribution to the payment.

LEMMA 4.1. *Fix any bidding strategies $\boldsymbol{b}$. For each auction $j$ and each $k \in [s]$, let*

$$q^{j,k} = \left( \frac{d_{j,k+1}}{\sum_{\ell \leq k} d_{j,\ell}}, \frac{\sum_{\ell < k} d_{j,\ell}}{\sum_{\ell \leq k} d_{j,\ell}} \right) \in \mathbb{R}_+^2.$$

*Moreover, let $\Delta^1 = \{(t_1, t_2) \in \mathbb{R}_+^2 \mid t_1 + t_2 = 1\}$ be the standard 1-simplex. Then, for each auction $j$, there exists $k \in [s]$, $r \in \Delta^1 \subseteq \mathbb{R}_+^2$, and $\alpha \in [0, 1]$, such that*

- *The total proxy value contributed by auction $j$ is at least*

$$\sum_{i \in [n]} \text{pval}_{i,j} \geq (\alpha \cdot q_1^{j,k} + (1 - \alpha) \cdot r_1) \cdot \text{opt}_j.$$

- *The total payment contributed by auction $j$ is at least*

$$\sum_{i \in [n]} p_{i,j} \geq (\alpha \cdot q_2^{j,k} + (1 - \alpha) \cdot r_2) \cdot \text{opt}_j.$$

Before diving into the proof, first let us understand the lemma. Intuitively, the lemma says that if (without loss of generality) the contribution of an auction $j$ to the optimal welfare is $\text{opt}_j = 1$, then $j$'s contributions to the total proxy value and the total payment are lower bounded by the two coordinates of $\alpha \cdot q^{j,k} + (1 - \alpha) \cdot r \in \mathbb{R}_+^2$ respectively. This point is a convex combination of $q^{j,k}$ and $r$. Observe that $\|q_{j,k}\|_1 \leq 1$ for all $j$ and $k$, so $q^{j,k}$ generally corresponds to the lossy component of the bound, whereas $r \in \Delta^1$ generally corresponds to the lossless component. Intuitively, the lossless component $r$ would not hurt the efficiency. Indeed, as we will see later, the lossy component $q^{j,k}$ dominates the PoA of the generalized second-price auction.

It may appear that the above lemma is unnecessarily complicated by the way it is presented (e.g., the use of points in $\mathbb{R}_+^2$), but we will see how this helps in the next part of the analysis where we aggregate the contributions. Also note that the lemma provides bounds on random variables, which is intentional. The rest of the subsection is devoted to the proof of the lemma.

PROOF OF LEMMA 4.1. Fix some auction $j$. Without loss of generality, suppose $i^*(j, k) = k$ for each $k \in [n]$, i.e., bidder $k$ has the $k$-th largest value.

*Warm-up: when bidder* 1 *would win the first slot.* We first consider the case where bidder 1 would win the first slot when bidding their true value. In this case, we argue that the contribution of auction $j$ to the value and the payment together is at least $\text{opt}_j$. This corresponds to $\alpha = 0$ in the statement of the lemma. We only need to show there exists $r \in \Delta^1$ that satisfies the conditions in the lemma. This is equivalent to the following:

$$\sum_{i \in [n]} \text{pval}_{i,j} + \sum_{i \in [n]} p_{i,j} \geq \text{opt}_j.$$

We give a charging argument that implies the above, which also serves as a warm-up for the more involved case that we will discuss later. Recall that $\text{opt}_j = \sum_{k \in [s]} v_{i^*(j,k),j} \cdot d_{j,k}$. Intuitively, in a charging argument, we would like to cover each $v_{i^*(j,k),j} \cdot d_{j,k}$ by two parts: the proxy value obtained by bidder $i^*(j, k)$, i.e., $\text{pval}_{i^*(j,k),j}$, and a fraction of total payment $\sum_{i \in [n]} p_{i,j}$ in auction $j$. The goal of a charging argument is to properly select $x_k$ for each $k$ such that $\text{pval}_{i^*(j,k),j} + x_k \geq v_{i^*(j,k),j} \cdot d_{j,k}$ while $\sum_k x_k$ not exceeding the total payment $\sum_{i \in [n]} p_{i,j}$. In particular, we will say "we charge $x_k$ to bidder $i^*(j, k)$".

Our charging scheme is illustrated in Figure 1. The idea is to differentiate the discount factors into "mass points", and consider the contribution to $\text{opt}_j$ of every mass point. For example, the top left mass point in each subfigure corresponds to $d_{j,1} - d_{j,2}$. The mass points in each row are equivalent to each other. We use the color of a mass point to represent the magnitude of the value or payment associated with this mass point before discounting. For example, the first column in the top left subfigure is black, which means the summation of the values associated with black points in the first column is $v_{1,j}$, the largest value in auction $j$. Similarly, $v_{2,j}$ and $v_{3,j}$ correspond to dark grey and light grey respectively. Since bidder 1 would win the first slot when they bid their true value, the proxy value of bidder 1 in auction $j$ is the contribution of this

column, which is

$$v_{1,j} \cdot [(d_{j,1} - d_{j,2}) + (d_{j,2} - d_{j,3}) + (d_{j,3} - d_{j,4}) + (d_{j,4} - d_{j,5}) + d_{j,5}] = v_{1,j} \cdot d_{j,1}.$$

This is the same as bidder 1's contribution to $\text{opt}_j$, which is always true in the case under discussion. As a result, we only need to bound the proxy values of other bidders.

To illustrate the idea of the charging scheme, assume there are only 3 bidders with positive values in auction $j$ (we will give a fully general argument later). Moreover, bidders 2 and 3 would win slots 3 and 5 when they bid their true values respectively. We show that the proxy value of bidders 2 and 3 in auction $j$ plus the payment made in auction $j$ is at least the contribution of bidders 2 and 3 to $\text{opt}_j$. Take bidder 2 for example. Bidder 2's proxy value is $v_{2,j} \cdot d_{j,3}$, and their contribution to $\text{opt}_j$ is $v_{2,j} \cdot d_{j,2}$, which means in order to cover bidder 2's contribution to $\text{opt}_j$, we need to charge $v_{2,j} \cdot (d_{j,2} - d_{j,3})$ from others' payment to bidder 2. Here, we need to pay special attention to two things: (1) we should never charge the same mass point to two bidders,[3] and (2) the payment associated with any mass point should never be less than the value of the bidder to which the mass point is charged. To guarantee (1), we divide the triangle of mass points into diagonal sequences, and charge only points in the $i$-th sequence from the top to bidder $i$, as illustrated in the top right subfigure. More specifically, we charge any mass point in the sequence that is equivalent to a mass point in the difference between the contribution to $\text{opt}_j$ and the proxy value of a bidder to that bidder.

To see why (2) is guaranteed, observe that any mass point in this difference must lie to the left of the column corresponding to the slot that the bidder would win, with at least one column in between separating the two. Again, take bidder 2 for example. As illustrated in the top right subfigure, the rightmost (and only) point in the difference within the second diagonal sequence is the second point in the first column in dark grey, whereas the column corresponding to the slot that bidder 2 would win is the third column, with the second column in between separating the two. We know that the second highest bidder under the fixed bidding strategy $\boldsymbol{b}$, whose slot corresponds to the second column, bids at least $v_{2,j}$. By the payment rule, this means the payment before discounting made by the bidder with the highest bid, whose slot corresponds to the first column, is at least $v_{2,j}$. So, property (2) holds when charging the second point in the first column to bidder 2. Similarly, one can check that the light grey points charged to bidder 3 also respect property (2). After performing the above charging, the total proxy value received by bidders 2 and 3 and the payment charged to the same bidders are illustrated in the bottom left subfigure. One can compare this to the contribution of these two bidders to $\text{opt}_j$ illustrated in the bottom right subfigure, and conclude that they are equivalent.

Now we present a fully general argument. For each bidder $i \in [s]$, we argue that the proxy value plus the payment charged to $i$ is at least $i$'s contribution to $\text{opt}_j$. Suppose the slot $i$ would win when bidding $v_{i,j}$ is $k$ (where $k = s + 1$ if $i$ would not win any of the $s$ slots). Consider the following cases:

---

[3]Note that we can still charge a mass point even if it already contributes to the proxy value of a bidder. This is not illustrated in the example in Figure 1.

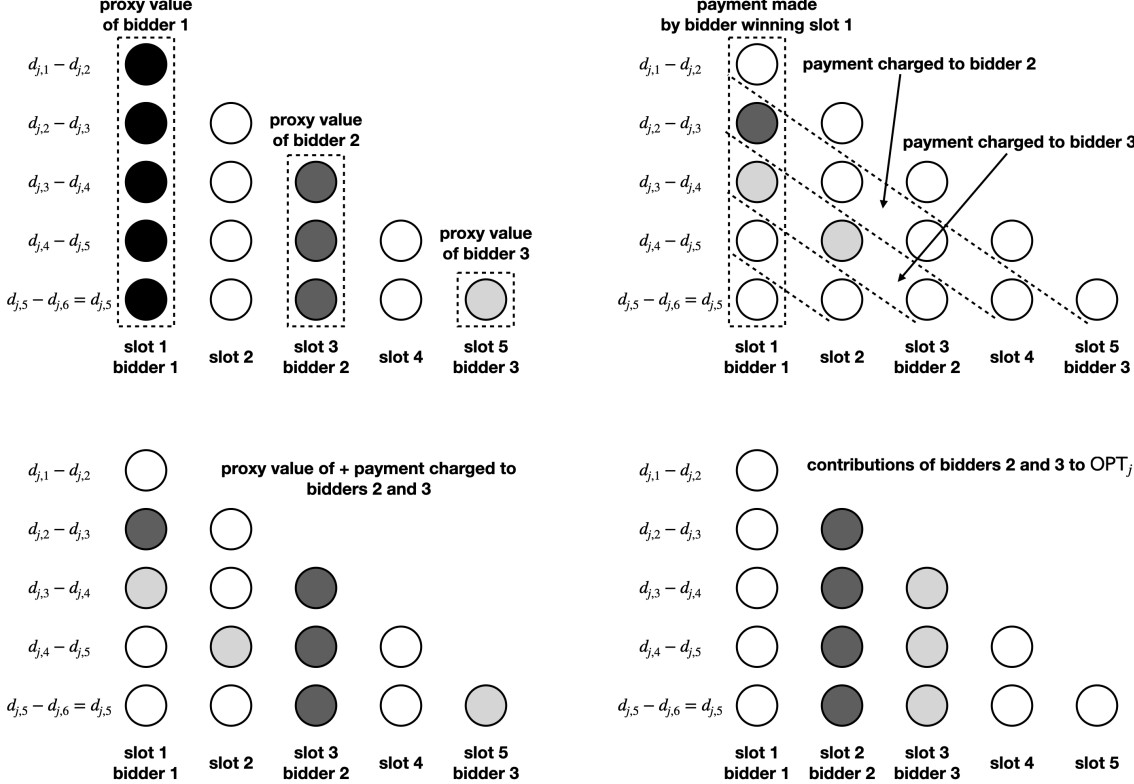

**Figure 1: Charging scheme used in the case where bidder $1$ would win the first slot.**

- If $i = 1$, then $k = 1$, and $\text{pval}_{i,j} = v_{1,j} \cdot d_{j,1}$ is precisely $i$'s contribution to $\text{opt}_j$.
- If $i > 1$ and $k \leq i$, then $\text{pval}_{i,j}$ is at least $i$'s contribution to $\text{opt}_j$, i.e.,

$$\text{pval}_{i,j} = v_{i,j} \cdot d_{j,k} \geq v_{i,j} \cdot d_{j,i}.$$

- If $i > 1$ and $k > i$, we need to charge at least $v_{i,j} \cdot (d_{j,i} - d_{j,k})$ to $i$ in terms of payment. The mass points we charge to $i$ are $\{(1, i), (2, i+1), \ldots, (k-i, k-1)\}$, where the two coordinates correspond to the indices of the column (from the left) and the row (from the top), respectively. Observe that the largest $k - 1$ bids are at least $i$'s value $v_{i,j}$; otherwise, $i$ would win a slot $< k$ using the phantom bidding strategy. Therefore, for any $\ell < k - 1$, the payment before discounting made by $i(j, \ell)$ has the following lower bound:

$$p_{i(j,\ell),j}/d_{j,\ell} = b_{i(j,\ell+1),j} \geq b_{i(j,k-1),j} \geq v_{i,j}.$$

So the total payment charged to $i$ is

$$\sum_{1 \leq \ell \leq k-i} (p_{i(j,\ell),j}/d_{j,\ell}) \cdot (d_{j,i-1+\ell} - d_{j,i+\ell})$$

$$\geq \sum_{1 \leq \ell \leq k-i} v_{i,j} \cdot (d_{j,i+1-\ell} - d_{j,i+2-\ell})$$

$$= v_{i,j} \cdot (d_{j,i} - d_{j,k}),$$

which is the difference to be filled between $i$'s contribution to $\text{opt}_j$ and $i$'s proxy value in $j$.

Finally, observe that no mass point is used twice across different bidders in the third case above. This means

$$\sum_{i \in [n]} \text{pval}_{i,j} + \sum_{i \in [n]} p_{i,j} \geq \text{opt}_j,$$

which establishes the lemma in the case where $v_{1,j} \geq b_{i(j,1),j}$.

*When bidder $1$ would not win the first slot.* Now consider the more challenging case where $v_{1,j} < b_{i(j,1),j}$, i.e., bidder 1's value is smaller than the highest bid. Suppose bidder 1 would win the $(k + 1)$-th slot if they bid their true value (where $k = s$ if bidder 1 would not win any of the $s$ slots). Then the largest $k$ bids are at least $v_{1,j}$. We will argue about the contribution of bidders $1, \ldots, k$ and the contribution of bidders $k + 1, \ldots, n$ separately, using different charging schemes. In particular, we will cover most of the contribution of the first $k$ bidders with highest values to $\text{opt}_j$ using the proxy value of bidder 1 and the payment made by the bidders who win the first $k - 1$ slots under $\boldsymbol{b}$. This corresponds to $q^{j,k}$ in the statement of the lemma. Then, for the other $n - k$ bidders, we reuse the charging scheme in the first case of the proof, and show that the proxy value of these bidders and the payment made by the bidders who win the other $s - k + 1$ slots under $\boldsymbol{b}$ together cover the full contribution of these $n - k$ bidders. This corresponds to $r$ in the statement of the lemma. The parameter $\alpha \in [0, 1]$ then captures how $\text{opt}_j$ is split between these two sets of bidders.

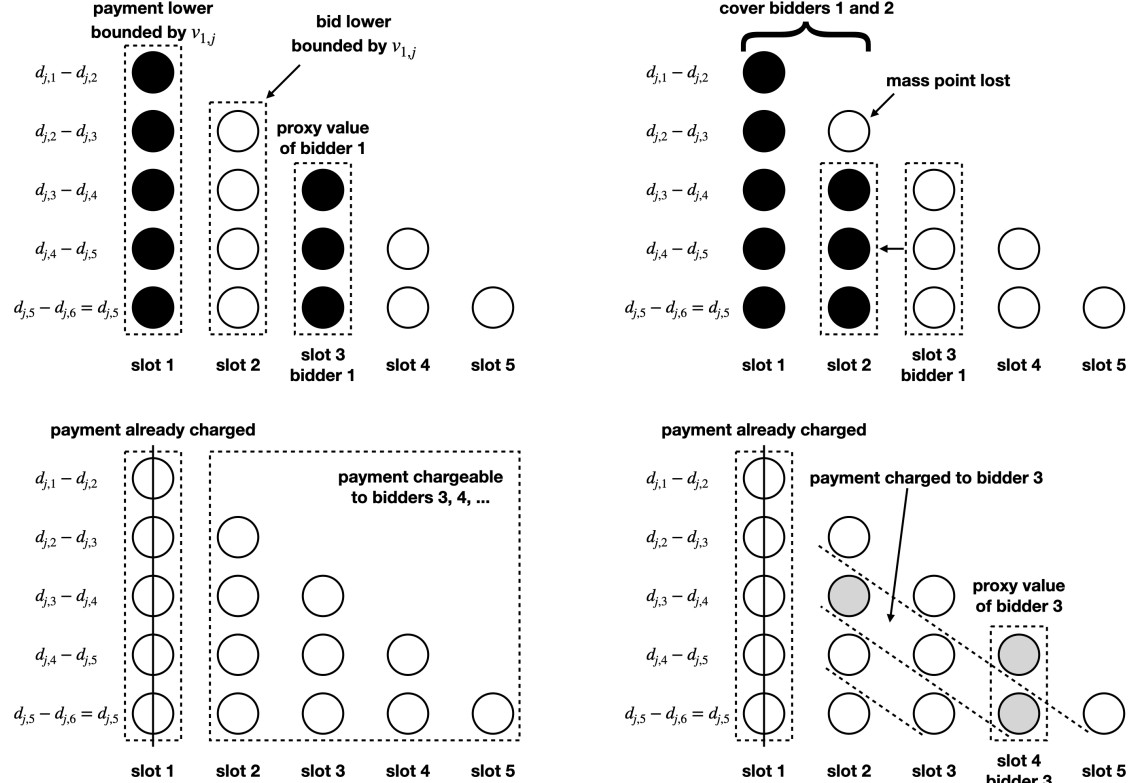

**Figure 2: Charging scheme used in the case where bidder 1 would not win the first slot.**

For the first $k$ bidders, we use a new charging scheme, which is depicted in Figure 2. Again, we first present the argument in an intuitive way using a simple example. Suppose $k = 2$, i.e., bidder 1 would win slot 3 when bidding $v_{1,j}$. As illustrated in the top left subfigure, this means two things: (1) the proxy value of bidder 1 is $v_{1,j} \cdot d_{j,3}$, and (2) the top 2 bids under $b$ (corresponding to slots 1 and 2) are at least $v_{1,j}$, so the top 1 payment (corresponding to slot 1) is at least $v_{1,j}$. To see how this almost covers the contribution of bidders 1 and 2 to $\text{opt}_j$, we shift the proxy value in the third column to the left, as shown in the top right subfigure. Then, the payment in the first column covers the full contribution of bidder 1. As for bidder 2, the worst-case situation is where $v_{2,j} = v_{1,j}$, in which case the black points in the second column covers the contribution of bidder 2 except for the top mass point in the column, which is inevitably lost in the charging scheme. So for the first 2 bidders, their total proxy value covers at least $d_{j,3}/(d_{j,1} + d_{j,2})$ of their contribution to $\text{opt}_j$, and the payment charged to them covers at least $d_{j,1}/(d_{j,1} + d_{j,2})$ of their contribution to $\text{opt}_j$. This corresponds to $q^{j,2}$ in the statement of the lemma.

Now we still need to cover the contribution of bidders 3, 4, and 5, and as depicted in the bottom left subfigure, the mass points that are still available for charging is everything except those in the first column, which were already charged when covering bidders 1 and 2. So we have a 4-by-4 triangle of mass points that can be used to cover 3 bidders, and we can reuse the charging scheme

introduced in the case where bidder 1 would win slot 1 to fully cover the contribution of these 3 bidders. This is illustrated in the bottom right subfigure. For example, if bidder 3 would win slot 4, then the proxy value of bidder 3 is $v_{3,j} \cdot d_{j,4}$, and we still need to charge a mass point that is worth $v_{3,j} \cdot (d_{j,3} - d_{j,4})$ to bidder 3 in terms of payment. For this we choose the second point in the second column, which has the right mass. Moreover, the payment before discounting corresponding to the second column is at least $v_{3,j}$, again because it is separated from the fourth column where bidder 3 would be. Similarly, we can charge other points to bidders 4 and 5 to cover all their contribution to $\text{opt}_j$.

Now we present a fully general argument. First consider bidders $1, \ldots, k$. The goal is to show that the total proxy value of these $k$ bidders is at least $d_{j,k+1}/\sum_{\ell \le k} d_{j,\ell}$ of their contribution to $\text{opt}_j$, and charge an amount of payment to these bidders that is at least $\sum_{\ell < k} d_{j,\ell}/\sum_{\ell \le k} d_{j,\ell}$ of their contribution to $\text{opt}_j$. For the first part, it suffices to consider the proxy value of bidder 1, which is

$$\text{pval}_{1,j} = v_{1,j} \cdot d_{j,k+1} = \frac{d_{j,k+1}}{\sum_{\ell \le k} d_{j,\ell}} \cdot \left( \sum_{i \le k} v_{1,j} \cdot d_{j,i} \right)$$

$$\ge \frac{d_{j,k+1}}{\sum_{\ell \le k} d_{j,\ell}} \cdot \left( \sum_{i \le k} v_{i,j} \cdot d_{j,i} \right),$$

where the inequality is because $v_{i,j} \le v_{1,j}$ for all $i \in [n]$. For the second part, we charge all the payment made by bidders who win

the first $k - 1$ slots under $\boldsymbol{b}$ to bidders $2, \ldots, k$. The amount we get is

$$
\begin{aligned}
\sum_{\ell \leq k-1} p_{i(j,\ell),j} &= \sum_{\ell \leq k-1} d_{j,\ell} \cdot b_{i(j,\ell+1),j} \geq \sum_{\ell \leq k-1} d_{j,\ell} \cdot b_{i(j,k),j} \\
&\geq \sum_{\ell \leq k-1} d_{j,\ell} \cdot v_{1,j} = \frac{\sum_{\ell < k} d_{j,\ell}}{\sum_{\ell \leq k} d_{j,\ell}} \cdot \sum_{i \leq k} v_{1,j} \cdot d_{j,i} \\
&\qquad \text{(bidder 1 would win slot } k+1 \text{ when bidding } v_{1,j}) \\
&\geq \frac{\sum_{\ell < k} d_{j,\ell}}{\sum_{\ell \leq k} d_{j,\ell}} \cdot \sum_{i \leq k} v_{i,j} \cdot d_{j,i}. \qquad (v_{1,j} \geq v_{i,j})
\end{aligned}
$$

This takes care of bidders $1, \ldots, k$. One can check that the argument goes through even if $k = s$, i.e., when bidder 1 would not win a slot when bidding $v_{1,j}$.

Now consider bidders $k + 1, \ldots, s$ (if $k < s$). The goal here is to show that the total proxy value of these $s - k$ bidders plus the total payment charged to them is at least their contribution to $\mathrm{opt}_j$. The argument is essentially the same as the one used in the first case of the proof, where bidder 1 would win slot 1 when bidding $v_{1,j}$. For each $i \in \{k + 1, \ldots, s\}$, let $k'$ be the slot $i$ would win when bidding $v_{i,j}$. Consider two cases:

- If $k' \leq i$, then $\mathrm{pval}_{i,j}$ is at least $i$'s contribution to $\mathrm{opt}_j$.
- If $k' > i$, we need to charge at least $v_{i,j} \cdot (d_{j,i} - d_{j,k'})$ to $i$ in terms of payment. The mass points we charge to $i$ are $\{(k, i), (k+1, i+1), \ldots, (k-1+k'-i, k'-1)\}$, where again the two coordinates correspond to the indices of the column (from the left) and the row (from the top), respectively. Observe that the largest $k' - 1$ bids are at least $i$'s value $v_{i,j}$, so for any $\ell < k' - 1$, the payment before discounting made by $i(j, \ell)$ has the following lower bound:

$$
p_{i(j,\ell),j}/d_{j,\ell} = b_{i(j,\ell+1),j} \geq b_{i(j,k'-1),j} \geq v_{i,j}.
$$

So the total payment charged to $i$ is

$$
\begin{aligned}
&\sum_{k \leq \ell \leq k-1+k'-i} (p_{i(j,\ell),j}/d_{j,\ell}) \cdot (d_{j,i-k+\ell} - d_{j,i-k+\ell+1}) \\
&\geq \sum_{k \leq \ell \leq k-1+k'-i} v_{i,j} \cdot (d_{j,i-k+\ell} - d_{j,i-k+\ell+1}) \\
&= v_{i,j} \cdot (d_{j,i} - d_{j,k'}),
\end{aligned}
$$

as desired.

Finally, observe that (1) the above charging scheme for bidders $k + 1, \ldots, s$ only uses mass points in columns $k, \ldots, s$, and (2) each mass point is used at most once across bidders $k + 1, \ldots s$. This means

$$
\sum_{k+1 \leq i \leq s} \mathrm{pval}_{i,j} + \sum_{k \leq \ell \leq s} p_{i(j,\ell),j} \geq \sum_{k+1 \leq i \leq s} v_{i,j} \cdot d_{j,i}.
$$

Now we put together the bounds for the two sets of bidders to conclude the proof. Let $\alpha$ be the unique number in $[0, 1]$ such that

$$
\sum_{i \leq k} v_{i,j} \cdot d_{j,i} = \alpha \cdot \mathrm{opt}_j, \qquad \sum_{k+1 \leq i \leq s} v_{i,j} \cdot d_{j,i} = (1 - \alpha) \cdot \mathrm{opt}_j.
$$

The bounds we have for bidders $1, \ldots, k$ now become

$$
\sum_{i \leq k} \mathrm{pval}_{i,j} \geq \alpha \cdot q_1^{j,k} \cdot \mathrm{opt}_j,
$$

$$
\sum_{\ell \leq k-1} p_{i(j,\ell),j} \geq \alpha \cdot q_2^{j,k} \cdot \mathrm{opt}_j.
$$

Moreover, the bounds we have for bidders $k + 1, \ldots, s$ imply the existence of $r \in \Delta^1$ such that

$$
\sum_{k+1 \leq i \leq s} \mathrm{pval}_{i,j} \geq (1 - \alpha) \cdot r_1 \cdot \mathrm{opt}_j,
$$

$$
\sum_{k \leq \ell \leq s} p_{i(j,\ell),j} \geq (1 - \alpha) \cdot r_2 \cdot \mathrm{opt}_j.
$$

Adding together the respective bounds for the proxy value and the payment, we get

$$
\sum_{i \in [n]} \mathrm{pval}_{i,j} \geq (\alpha \cdot q_1^{j,k} + (1 - \alpha) \cdot r_1) \cdot \mathrm{opt}_j,
$$

$$
\sum_{i \in [n]} p_{i,j} \geq (\alpha \cdot q_2^{j,k} + (1 - \alpha) \cdot r_2) \cdot \mathrm{opt}_j.
$$

This concludes the proof. □

## 5 AGGREGATING THE CONTRIBUTIONS OF ALL AUCTIONS

Lemma 4.1 bounds the contributions of each auction to the total proxy value and the total payment. In order to obtain a PoA bound, we need to aggregate the bounds provided by Lemma 4.1 over all auctions in a worst-case fashion. This subsection is devoted to such an aggregation argument. By the end of the subsection, we will have proved the main result of the paper, stated below.

THEOREM 5.1. *For any $m, s$, and $\{d_{j,k}\}$, let $j_0 = \mathrm{argmin}_{j \in [m]} \frac{d_{j,2}}{d_{j,1}}$. Then*

$$
\mathrm{PoA}(m, s, \{d_{j,k}\}) \geq
$$

$$
\min_{j \in [m], k \in \{2, \ldots, s\}} \frac{d_{j_0,2} \cdot \sum_{\ell < k} d_{j,\ell}}{d_{j_0,1} \cdot \sum_{\ell < k} d_{j,\ell} - d_{j_0,1} \cdot d_{j,k+1} + d_{j_0,2} \cdot \sum_{\ell \leq k} d_{j,\ell}}.
$$

The above bound inevitably has a complex form. Nevertheless, as discussed in the intial part of the proof, it has an intuitive geometric interpretation: The bound is the $x$- or $y$-coordinate (they are the same) of the lower intersection of the convex closure of $\{q^{j,k}\}$ and the line $x = y$. Later we will also present a simplified (but looser) version of the bound in Corollary 5.2. We defer the proof of Theorem 5.1 to the appendix due to space constraints.

*A simplified bound.* Now we present a simplified bound on the PoA of the generalized second-price auction, which can be obtained by considering the sum of the contributions to the value and the payment as in Lemma 4.1. This bound can be viewed as an approximation of Theorem 5.1. Geometrically, as we will see, the way we derive this bound is equivalent to considering a tangent line of the convex closure of $\{q^{j,k}\}$. Conceptually, this simplified bound highlights the qualitative relation between the smoothness of the discount factors and the efficiency of the generalized second-price auction.

Corollary 5.2. *For any $m$, $s$, and $\{d_{j,k}\}$,*

$$\text{PoA}(m, s, \{d_{j,k}\}) \geq \min_{j \in [m], k \in [s]} \frac{1}{2} \cdot \frac{\sum_{\ell < k} d_{j,\ell} + d_{j,k+1}}{\sum_{\ell \leq k} d_{j,\ell}}.$$

Proof. The statement can be derived from Theorem 5.1 by relaxing the bound therein. Here we present a more intuitive geometric argument that relies on Lemma 4.1. Recall that the bound in Theorem 5.1 corresponds to the lower intersection of the convex closure of $\{q^{j,k}\}$ and the line $x = y$. This can be relaxed to the intersection of the tangent line of the convex closure that is parallel to $x + y = 1$, and the line $x = y$. Below we argue that the latter intersection corresponds to the bound in the statement to be proved.

Observe that the tangent line that we care about must be induced by the point in $\{q^{j,k}\}$ with the smallest sum of the two coordinates. Moreover, the intersection between this tangent line and $x = y$ must have the same sum of the two coordinates, which means each coordinate of the intersection is precisely half of the sum. Recall that the sum of the two coordinates of each $q^{j,k}$ is $\frac{\sum_{\ell < k} d_{j,\ell} + d_{j,k+1}}{\sum_{\ell \leq k} d_{j,\ell}}$. Taking the minimum over $j$ and $k$ and dividing it by two gives the bound to be proved. □

Note that the simplified bound can never be actually attained: In order for the simplified bound to be equal to the bound in Theorem 5.1, it has to be the case that the lower intersection of the convex closure and $x = y$ is some point in $\{q^{j,k}\}$, say $q^{j^*,k^*}$. Then we know that $q_1^{j^*,k^*} = q_2^{j^*,k^*}$, which can only happen when $k^* = 2$ and $d_{j^*,1} = d_{j^*,2} = d_{j^*,3}$. This means $q_1^{j^*,k^*} = q_2^{j^*,k^*} = 1/2$, and the simplified bound would be $1/2$. However, the latter is impossible, because the PoA of the generalized second-price auction is always strictly worse than $1/2$. This also highlights the necessity of our aggregation argument used in the proof of Theorem 5.1 for establishing a tight PoA bound.

## 6 TIGHTNESS OF THE BOUND

Now we discuss the tightness of the bound. The notion of tightness we consider is a worst-in-class one: We show that among all instances that share the same bound as given by Theorem 5.1, there is one for which the bound is tight.

Theorem 6.1. *For any $t \in (0, \frac{1}{2})$, there exists a positive integer $s \geq 2$ and a real number $x \in \mathbb{R}_+$, such that if we set $m = s$, $d_{j,1} = x+1$ for each $j \in [m]$, and $d_{j,k} = 1$ for each $j \in [m]$ and $k > 1$, then*

- *$\text{PoA}(m, s, \{d_{j,k}\}) = t$, and*
- *$\frac{s-1+x}{(1+x) \cdot (s-1+x)+s+x} = t$.*

*In particular, the quantity in the second condition is the bound given in Theorem 5.1 evaluated on the instance $(m, s, \{d_{i,j}\})$.*

Proof. We first pick $s$ and $x$ satisfying the second condition. Observe two facts:

- When $x = 0$, the quantity reduces to $\frac{s-1}{2s-1}$, which increases as $s$ increases and goes to $1/2$.
- Fixing any $s$, when $x$ goes to $\infty$, the quantity goes to $0$.

So one way to pick $(s, x)$ is to let $s$ be any positive integer such that $\frac{s-1}{2s-1} \geq t$, and there must exist $x$ such that $(s, x)$ satisfies the second condition.

Suppose without loss of generality ties are broken in favor of bidders with smaller indices. We construct a valuation profile $\{v_{i,j}\}$ with $n = 2s$ bidders and deterministic bidding strategies $\boldsymbol{b}$ that form an equilibrium, parametrized by some $\varepsilon > 0$, where

$$\frac{\sum_{i,j} \text{val}_{i,j}}{\sum_j \text{opt}_j} \to t, \text{ as } \varepsilon \to 0.$$

We partition the $n = 2s$ bidders into two groups each of size $s$, $\{1, \ldots, s\}$ and $\{s+1, \ldots, 2s\}$. Bidder 1 has value $(1+\varepsilon) \cdot (1+x)$ in auction 1, bidder $s$ has value $\varepsilon$ in auction 1, while all other bidders have value 0 in auction 1. For each $i \in \{2, \ldots, s-1\}$, bidder $i$ has value $(1+\varepsilon)$ in auction $i$, bidder $s$ has value $\varepsilon$ in auction $i$, while all other bidders have value 0 in auction $i$. This fully specifies the valuation profile in the first $s-1$ auctions. As for auction $s$, each bidder $i \in \{s+1, \ldots, 2s\}$ in the second group has value 1, while all bidders in the first group have value 0.

One equilibrium is where for each $i \in [s-1]$, bidder $i$ bids $b_{i,i} = \varepsilon$ in auction $i$, bidder $s$ bids $\infty$ in auction $i$, while all other bidders $i'$ bid $b_{i',i} = 0$ in auction $i$. As a result, bidder $s$ wins the first slot in each of these $s-1$ auctions, whereas bidder $i$ wins the second slot. So bidder $s$ receives total value $\varepsilon \cdot (1+x) \cdot (s-1)$ and pays the same amount in total, bidder 1 receives value $(1+\varepsilon) \cdot (1+x)$ in auction 1 and pays 0, and each bidder $i \in \{2, \ldots, s-1\}$ receives value $1+\varepsilon$ and pays 0 in auction $i$. As for auction $s$, each bidder $i \in [s]$ in the first group bids $b_{i,m} = 1+\varepsilon$, while each bidder $i' \in \{s+1, \ldots, 2s\}$ in the second group bids $b_{i',m} = 0$. As a result, no bidder receives any value in auction $s$, bidder 1 pays $(1+\varepsilon) \cdot (1+x)$, and each bidder $i \in [s-1]$ pays $1+\varepsilon$. One can check this is in fact an equilibrium, since (1) the ROI constraint of each bidder is satisfied, (2) no bidder $i \in [s-1]$ can win the first slot in auction $i$ without violating their ROI constraint, and (3) no bidder in the second group can win anything in auction $s$ without violating their ROI constraint.

Now we compute the ratio between the welfare in equilibrium and the optimal welfare. The optimal welfare is simply

$$\sum_j \text{opt}_j = \text{opt}_1 + (s-2) \cdot \text{opt}_2 + \text{opt}_s$$

$$= [(1+\varepsilon) \cdot (1+x) \cdot (1+x) + \varepsilon]$$
$$+ (s-2) \cdot [(1+\varepsilon) \cdot (1+x) + \varepsilon] + [(1+x) + s - 1]$$
$$= (1+x) \cdot (s-1+x) + s + x + O(\varepsilon),$$

where $O(\cdot)$ hides a constant that depends on $s$ and $x$. On the other hand, the welfare in equilibrium is

$$\sum_{i,j} \text{val}_{i,j} = \text{val}_{1,1} + (s-2) \cdot \text{val}_{i,i} + (s-1) \cdot \text{val}_{s,1}$$

$$= (1+x) \cdot (1+\varepsilon) + (s-2) \cdot (1+\varepsilon) + (s-1) \cdot \varepsilon \cdot (1+x)$$
$$= s - 1 + x + O(\varepsilon),$$

where again $O(\cdot)$ hides a constant that depends on $s$ and $x$. So the ratio between the two goes to $t = \frac{s-1+x}{(1+x) \cdot (s-1+x)+s+x}$ as $\varepsilon$ goes to 0. This concludes the proof. □

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

## A  PROOF OF THEOREM 5.1

In this appendix we prove Theorem 5.1.

PROOF OF THEOREM 5.1. Again we start with an intuitive geometric interpretation in Figure 3, which also serves as an overview of the proof. For simplicity, assume $m = 2$ and $s = 3$. Lemma 4.1 is visualized in the top left subfigure for auction 1. The dashed lines connect points (which correspond to $q^{1,k}$ for $k \in [s]$) on the Pareto frontier of auction 1, and the solid line $x + y = 1$ corresponds to the lossless component $r$. The bounds given in Lemma 4.1 can only be

points within the region bounded by the dashed lines, $x + y = 1$, and the axes, as illustrated (some points within the region do not correspond to bounds in Lemma 4.1). Note that everything is normalized by the contribution of auction 1, $\mathrm{opt}_1$, in this subfigure.

Now in the top right subfigure, we further superimpose the Pareto frontier of auction 2, and consider the aggregated contribution of auctions 1 and 2. Here, we view the respective contributions of auctions 1 and 2 as points above the respective Pareto frontiers, weighted by $\mathrm{opt}_1$ and $\mathrm{opt}_2$. Then, the aggregated contribution of the two auctions, normalized by $\mathrm{opt}_1 + \mathrm{opt}_2$, is the weighted average of these two points. Since the optimal welfare can be arbitrarily distributed between the two auctions, the aggregated point can be any convex combination of any two points in the respective regions. In other words, the aggregated contribution of the two auctions can be any point in the region bounded by solid lines, as illustrated.

To determine the worst-case aggregated bounds, consider the bottom left subfigure. Recall that in expectation, the welfare in equilibrium is lower bounded by both the total proxy value and the total payment after aggregation (we will give a detailed argument later). That is, both coorinates of the point corresponding to the aggregated bounds are lower bounds of the welfare in equilibrium, and the worst case is where the larger one of the two coordinates is minimized. Now since the region bounding the aggregated point is convex (it is the convex closure of the Pareto frontiers), the worst case must be achieved when the two coordinates are equal. Geometrically, this means the worst-case point is the intersection of the lower envelope of $\{q^{j,k}\}$ and the line $x = y$, as illustrated in the bottom left subfigure. This already gives a way for computing the lower bound on the worst-case welfare in equilibrium.

To further simplify the bound, observe that the only points on the right of the line $x = y$ are $\{q^{j,1}\}_{j \in [m]}$. This is because for any $j$ and $k \geq 2$,

$$\sum_{\ell < k} d_{j,\ell} \geq d_{k+1,\ell},$$

which means the $y$-coordinate of the point is no smaller than the $x$-coordinate. So, the intersection corresponding to the worst-case bounds must be induced by some point from $\{q^{j,1}\}_{j \in [m]}$, and some point from the rest of the Pareto frontiers. Moreover, the $y$-coordinate of each $q^{j,1}$ is 0, so there is a worst point among $\{q^{j,1}\}$ regardless of which other point we pick, and the worst point is the one with the smallest $x$-coordinate. Once we fix this worst point, we only need to consider each point in the rest of the Pareto frontiers and find the one inducing the worst-case bounds in combination with the fixed point. This is illustrated in the bottom right subfigure.

Now we give a fully general proof. To aggregate Lemma 4.1 over auctions, we will show that there exist weights $w_{j,k}$ for each $j \in [m]$ and $k \in [s]$ where $\sum_{j,k} w_{j,k} = 1$, such that

$$\sum_{i,j} \mathrm{pval}_{i,j} \geq q_1 \cdot \sum_j \mathrm{opt}_j, \qquad \sum_{i,j} p_{i,j} \geq q_2 \cdot \sum_j \mathrm{opt}_j$$

where $q = \sum_{i,j} w_{j,k} \cdot q^{j,k}$. Here we closely follow the plan illustrated in Figure 3. By Lemma 4.1, for each $j$, there exists $k_1$ and $k_2 \in [s]$,

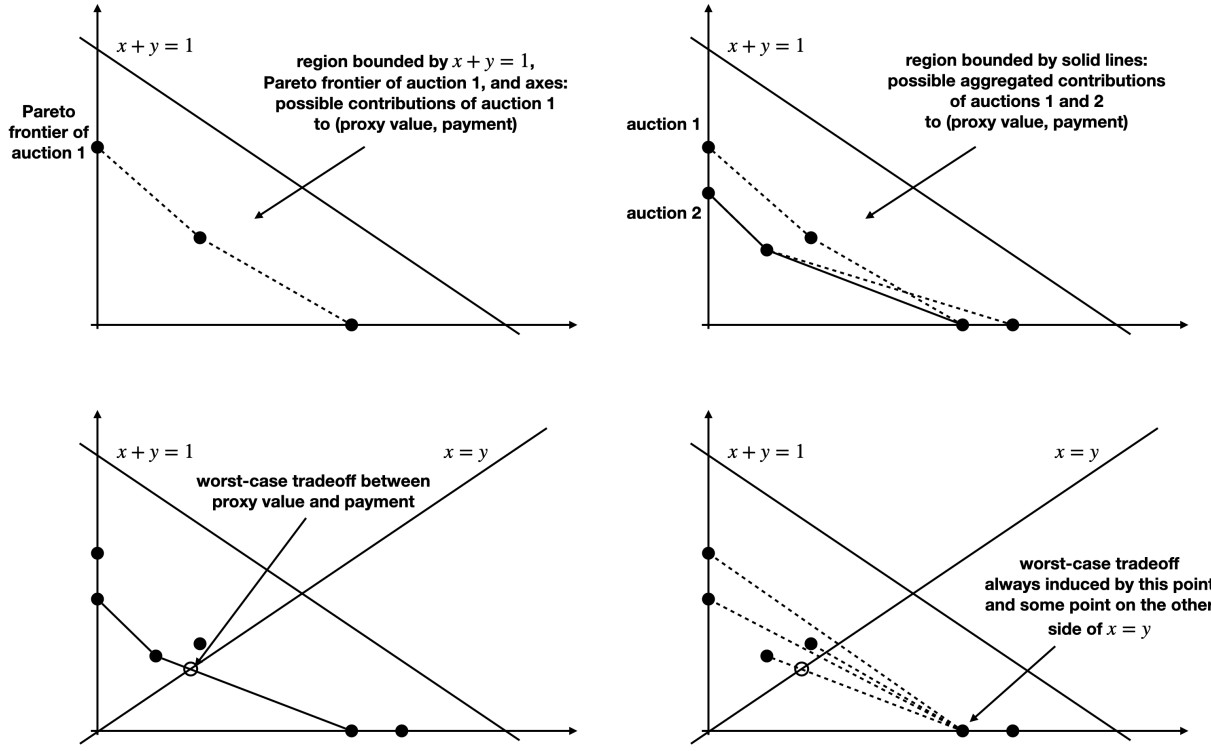

**Figure 3: Geometric interpretation of the aggregation argument.**

and $w_1$ and $w_2 \in \mathbb{R}_+$ where $w_1 + w_2 = 1$, such that

$$\sum_i \mathsf{pval}_{i,j} \geq (w_1 \cdot q^{j,k_1} + w_2 \cdot q^{j,k_2})_1 \cdot \mathsf{opt}_j,$$

$$\sum_i p_{i,j} \geq (w_1 \cdot q^{j,k_1} + w_2 \cdot q^{j,k_2})_2 \cdot \mathsf{opt}_j.$$

In particular, this is because the lower envelope of $\{q^{j,k}\}_k$ is below $\Delta_1$. So for $j$, we let

$$w_{j,k_1} = w_1 \cdot \frac{\mathsf{opt}_j}{\sum_{j'} \mathsf{opt}_{j'}}, \qquad w_{j,k_2} = w_2 \cdot \frac{\mathsf{opt}_j}{\sum_{j'} \mathsf{opt}_{j'}},$$

and $w_{j,k} = 0$ for $k \in [s] \setminus \{k_1, k_2\}$. One can check $\{w_{j,k}\}$ satisfy the condition above.

The existence of these weights means that the lower bounds normalized by the optimal welfare is a point in the convex closure of $\{q^{j,k}\}$. Given this, we consider the worst-case point and the corresponding lower bound of the welfare. As illustrated in Figure 3, the worst-case point is the intersection of the lower envelope of $\{q^{j,k}\}$ with the line $x = y$. Moreover, as discussed earlier, since $\{q^{j,1}\}$ are the only points on the right of $x = y$, we only need to consider segments between pairs of points in $\{q^{j,1}\} \times \{q^{j,k}\}_{k>1}$. And since $q_2^{j,1} = 0$ for all $j$, we can further restrict to the leftmost point in $\{q^{j,1}\}$, whose index is $j_0 = \operatorname{argmin}_j \frac{d_{j,2}}{d_{j,1}}$. For each $j$ and $k > 1$, one can then compute the $x$- or $y$-coordinate (they are the same) of the intersection of $x = y$ and the line determined by $q^{j_0,1}$

and $q^{j,k}$, which is

$$\frac{d_{j_0,2} \cdot \sum_{\ell < k} d_{j,\ell}}{d_{j_0,1} \cdot \sum_{\ell < k} d_{j,\ell} - d_{j_0,1} \cdot d_{j,k+1} + d_{j_0,2} \cdot \sum_{\ell \leq k} d_{j,\ell}}.$$

Taking the minimum over $j$ and $k > 1$, we get

$$\frac{\max \left\{ \sum_{i,j} \mathsf{pval}_{i,j}, \sum_{i,j} p_{i,j} \right\}}{\sum_j \mathsf{opt}_j} \geq$$

$$\min_{j \in [m], k \in \{2,\dots,s\}} \frac{d_{j_0,2} \cdot \sum_{\ell < k} d_{j,\ell}}{d_{j_0,1} \cdot \sum_{\ell < k} d_{j,\ell} - d_{j_0,1} \cdot d_{j,k+1} + d_{j_0,2} \cdot \sum_{\ell \leq k} d_{j,\ell}}.$$

The only step left is to relax the left hand side of the above to the PoA. To this end, we only need to show that in equilibrium,

$$\sum_{i,j} \mathbb{E}[\mathsf{val}_{i,j}] \geq \sum_{i,j} \mathbb{E}[\mathsf{pval}_{i,j}], \qquad \sum_{i,j} \mathbb{E}[\mathsf{val}_{i,j}] \geq \sum_{i,j} \mathbb{E}[p_{i,j}].$$

The latter follows directly from each bidder $i$'s ROI constraint. As for the former, since $\boldsymbol{b}$ consist of equilibrium strategies, fixing $\boldsymbol{b}_{-i}$, each bidder $i$ must be maximizing their expected total value subject to the ROI constraint. On the other hand, the ROI constraint is satisfied under the proxy bidding strategy $b_{i,j} = v_{i,j}$, so the expected proxy value cannot exceed the actual expected value in equilibrium. That is, for each $i \in [n]$,

$$\sum_j \mathbb{E}[\mathsf{val}_{i,j}] \geq \sum_j \mathbb{E}[\mathsf{pval}_{i,j}].$$

Summing over $i$ gives the desired bound. This concludes the proof of the theorem. □

