# OpenReview forum: "Efficiency of the Generalized Second-Price Auction for Value Maximizers"
_ACM.org/TheWebConf/2024/Conference — TheWebConf24_

### Official Review · Reviewer_NM9o · 2023-11-18

**Novelty:** 5
**Technical Quality:** 5

**Review:**

Summary:
The paper investigates the price of anarchy (PoA) in the context of the generalized second-price auction (GSP) where bidders act as value maximizers (autobidders). The authors find that the PoA can be as bad as 0 in the worst case, contrasting with the 1/2 PoA of running the VCG mechanism in autobidding scenarios. The paper provides a fine-grained analysis of PoA concerning discount factors in GSP, emphasizing the relationship between the smoothness of discount factors and the efficiency of the auction.

Strengths:
1.	Original Contribution: The paper makes a good contribution by characterizing the PoA of GSP auctions for value maximizers.
2.	Fine-Grained Analysis: The fine-grained analysis of PoA with respect to discount factors adds depth to the understanding of the efficiency of the GSP auction.
3.	Tight Bounds: The paper establishes tight PoA bounds in the worst-in-class sense, providing a nuanced perspective on the performance of GSP auctions.

Weaknesses:
1.	Complexity: The paper acknowledges the complexity of the bounds provided, which might limit accessibility for readers not deeply familiar with the subject matter. Consider simplifying the presentation or providing additional explanations to enhance clarity.
2.	Practical Relevance: While the worst-case scenario with a PoA of 0 is discussed, the paper notes its impracticality in real-world auctions. However, it would be beneficial to elaborate on the practical relevance of the findings and discuss potential real-world implications or applications.

**Questions:**

Can the authors provide a more intuitive interpretation of the practical implications of the assumption “suppose all bidders bid deterministically, and never bid below their values” in Section 3? How likely is such a scenario in realistic auction settings?

**Reviewer Confidence:**

2: The reviewer is willing to defend the evaluation, but it is likely that the reviewer did not understand parts of the paper

**Scope:**

4: The work is relevant to the Web and to the track, and is of broad interest to the community

---

### Official Review · Reviewer_NbQH · 2023-11-21

**Novelty:** 4
**Technical Quality:** 5

**Review:**

The authors consider the problem of finding the price of anarchy of the generalized second price
auction when bidders are value maximizers. In an autobidding context (which are becoming
more prevalent as advertising auctions become more ubiquitous on the web), bidders (i.e.,
advertisers) specify constraints to a tool which then automatically plans bids for the bidder
aiming to satisfy these constraints and maximize the resulting value enjoyed by the bidder
across a series of auctions. These constraints typically come in two main flavors - namely hard
budget constraints and return-on-spend constraints. This paper investigates the well-known
generalized second-price auction (a non-truthful extension of the second price auction to
polymatroid constraints like those of ad auctions) in the presence of value maximizing bidders
who have only return-on-spend constraints and studies the resulting price of anarchy of the
setting (i.e., the worst-case ratio across all instances of the welfare obtained in equilibrium
versus the optimal welfare).

The central result of this paper is a tight analysis of the price of anarchy of the GSP auction in
presence of value-maximizing bidders with ROS constraints. In particular, the authors
demonstrate a bound that, essentially, depends on the steepness of the "drop-off’’ in
clickthrough probabilities between the first and second advertisement slots. As the ratio of the
second slot clickthrough rate over the first slot clickthrough rate approaches 0, the price of
anarchy approaches 0 as well. This feels, in spirit, similar to the the well-known fact that the
second-price auction for a single item with utility maximizing bidders has a price of anarchy of 0.
The authors achieve their parameterized lower bound on the price of anarchy via a charging
argument to bound the contribution to the obtained welfare of each individual auction and a
geometric argument to find a worst-case aggregation of the welfares across auctions. They
then complement this lower bound with a matching upper bound.

On the positive side, this paper has a few nice features. First, it contributes to an area of
growing research activity, namely, auctions "in the autobidding world’’, which is likely to be of
interest to many parties at WebConf. Second, the paper takes a well-known auction which in
the past was widely used, i.e., the generalized second price auction, and provides tight
guarantees for its price of anarchy in the presence of value-maximizers with expected return on
spend constraints. Finally, the analysis is mostly easy to follow and presented in a
straightforward manner.

On the other hand, there are aspects of this paper which, in my view, are less
interesting/exciting. First, the fact that the price of anarchy of GSP approaches zero in this
setting is not really surprising. Indeed, as mentioned before and as cited in the paper, the
second-price auction has a price of anarchy of 0 even in the simple case of two bidders and a
single item if the bidders are quasi-linear utility maximizers. The authors begin to address this
by pointing out that VCG (the extension of the second-price auction to general settings)
achieves price of anarchy 1/2 when bidders do not adopt dominated strategies, i.e., bidders
cannot "underbid’’ their value, whereas underbidding is not a dominated strategy in GSP.
Whether or not this is the only difference between the two auctions causing the disparity in
guarantee (i.e., underbidding is what leads to the poorer performance of GSP) should probably
be addressed in detail in the paper. More concretely, since the "bad’’ equilibria of the second
price auction result from underbidding, ruling out underbidding in VCG but not GSP and then
arguing that GSP performs worse is not a particularly surprising takeaway message.
Second, while the analysis is presented mostly cleanly, the actual analysis itself is not, in my
view, particularly "difficult’’ or likely to lead to insights elsewhere. While the authors claim that
their "high-level approach can be adapted to other auction formats with local changes’’, it is not

immediately clear that the changes necessary to analyze a fixed auction of interest are small.
Adding some evidence to this effect would strengthen the technical message of the paper.
On the whole, while I think that this paper has some good qualities, I have some reservations
about the overall picture and whether the results and analyses herein are of significant enough
interest. I outline some smaller comments regarding presentation and/or typographical issues
below.

Smaller comments

While I appreciate that the authors wanted to give intuition for the proof via a diagram and high-
level discussion, I found the high level discussion no easier (actually slightly harder) to digest
than the analysis itself. For example the explanation of "shifting’’ the proxy value on page 6
beginning at line 620 was not particularly illuminating to me. It seems much more direct to
argue that $v_{1,j} \geq v_{2,j}$ and as such, the total contribution in the optimal solution of
$v_{1,j}$ and $v_{2,j}$ is no more than $v_{1,j}\cdot(d_{j,3} + d_{j,2})$ and you obtain
$d_{j,3}v_{1,j}$ as proxy value and $d_{j,1}v_{1,j}$ in payment. I think there is value to
including the figures and intuition, but I would suggest taking a second pass to see if the
intuition can be presented in a clearer, more concise way.

Lines 411-412: “which is always true in the case under discussion” — I’m not sure what this
means here. Perhaps it’s adding something of value, but the wording is unclear to me.

Lines 454-455: “and conclude that they are equivalent” — Are these two quantities equivalent?
It seems that you have an inequality (albeit, in the direction you need).

Lines 681-682: I would suggest you remind readers what $k$ is at this point, as this feels like
the start of the main argument.

[After rebuttal] Thank you for your responses to my questions and the questions of the other reviewers.  I do think that the technical takeaways from this paper are less exciting given the fact that the bad price of anarchy is, in a sense, solely due to underbidding, but I agree with the authors that no-underbidding is not easy to justify in the GSP.

**Questions:**

Does the GSP have no-underbidding equilibria with bad price of anarchy?  If so, this would suggest that there is something fundamentally different happening in GSP than VCG.

Do your have any examples of auctions where your analytical tools transfer with limited "local" changes and show price of anarchy bounds?

**Reviewer Confidence:**

3: The reviewer is confident but not certain that the evaluation is correct

**Scope:**

4: The work is relevant to the Web and to the track, and is of broad interest to the community

---

### Official Review · Reviewer_da3D · 2023-11-23

**Novelty:** 5
**Technical Quality:** 6

**Review:**

**Summary of the paper**

(topic) This paper studies the efficiency of generalized second-price auction when bidders are auto-bidders, measured by the price of anarchy (PoA).

(problem setting) The problem setting considers multi-auction model with n bidders and m auctions, where each auction has s(s<n) winning slots. In each auction j, each bidder i has a value v_{i, j} and each slot k has a discount factor d_{j, k}. The values of the discount factors are defined as d_{j, k} >= d{j, k+1} for k \in [s] and d_{j, k} = 0 for k > s.
In the generalized second-price auction, each bidder submits a bid d_{i, j} in each auction j. All bids are ranked afterward and the bidder with k_th largest bid pays the (k+1)_th bid multiplying kth slot’s discount, while receives the value of v_{i, j} * d{j, k}.
The auto-bidders are ROI-constrained value-maximizers. They aim at maximizing their total value they receive with the constraint that their total payment should be no larger than the total value.

(results) The paper studies the worst-case efficiency of the generalized second-price auction in equilibrium as is measured by PoA. The PoA is defined as the ratio between the total value received by all bidders in the worst equilibrium and the optimal welfare.
The authors bound PoA based on the discount factors, show that in the worst case the PoA can be 0. Except for the tight but complex bound, they also provide a slightly loose but simplified bounds showing that the PoA is better with smoother discount factors.

(techniques) The bounds are obtained in two steps: first, establish the Pareto frontier of welfare decomposition between value and payment for each auction, and second, aggregates the Pareto frontiers across auctions in an optimal way.

**Evaluation**

My research area is not in auction, so I am uncertain about the significance of the results in this paper. It seems to me that bounding the PoA is one of the fundamental problems in auction theory, and most problems related to PoA in the auction literature are technical challenging. Therefore, it seems to me that this paper provides a reasonable contribution. In addition, I found the model studied in this paper to be well-motivated.

**Questions:**

Minor question:
The quote in Line 257 “[Deng et al. 2021]” should be “[Deng et al. 2021a]“ or “[Deng et al. 2021b]“?

**Ethics Review Description:**

N.A.

**Reviewer Confidence:**

2: The reviewer is willing to defend the evaluation, but it is likely that the reviewer did not understand parts of the paper

**Scope:**

3: The work is somewhat relevant to the Web and to the track, and is of narrow interest to a sub-community

---

### Official Review · Reviewer_saBJ · 2023-11-25

**Novelty:** 6
**Technical Quality:** 6

**Review:**

The paper provides Price of Anarchy results for GSP auctions when agents are not utility-maximizers, but value-maximizers (i.e. they do not care on how much they pay). This is indeed the case in many ad auctions, in which people usually describe a budget, and automated agents  aim to maximize the value of achieved slots subject to do not spend more than the budget.

The paper provide a precise bound depending on the relationship among discount factors of different slots. The paper also proves that the bound is tight.

The problem of computing the PoA of non-strategy proof auctions is a well-studied problem, and this paper fills the gap by considering value-maximizers problems, that, as suggested above, is even more relevant to the current state of Web. The paper is well-written and also technically deep. In my opinion, it deserves acceptance.

**Questions:**

None

**Reviewer Confidence:**

3: The reviewer is confident but not certain that the evaluation is correct

**Scope:**

4: The work is relevant to the Web and to the track, and is of broad interest to the community

---

### Decision · Program_Chairs · 2024-01-22

**Decision:**

Accept

**Comment:**

The paper studies the price of anarchy in repeated GSP auctions where bidders are value maximizers (i.e. they maximize total expected value, subject to total expected cost not exceeding total expected value). They analyze the price of anarchy of such settings, parameterized by the discount factors used for the slots in GSP.

 The reviewers appreciated the relevance of the question that the authors studied, and while the results were perhaps not found to be too surprising, the analysis was non-trivial, yielding an overall valuable addition to the literature. As such the paper is recommended for acceptance.